# A retrospective multicenter comparison of conditional cancer-specific survival between laparoscopic and open radical nephroureterectomy in locally advanced upper tract urothelial carcinoma

**Sung Han Kim**[1☉], **Mi Kyung Song**[2☉], **Ja Hyeon Ku**[3], **Seok Ho Kang**[4], **Byong Chang Jeong**[5], **Bumsik Hong**[6]*, **Ho Kyung Seo**[7]*

1 Department of Urology, Center for Urologic Cancer, National Cancer Center, Goyang, Republic of Korea,
2 Health Insurance Research Institute, National Health Insurance Service, Wonju, Republic of Korea,
3 Department of Urology, Seoul National University Hospital, Seoul National University College of Medicine, Seoul, Republic of Korea, 4 Department of Urology, Korea University Anam Hospital, Korea University College of Medicine, Seoul, Republic of Korea, 5 Department of Urology, Samsung Medical Center, Sungkyunkwan University School of Medicine, Seoul, Republic of Korea, 6 Department of Urology, Asan Medical Center, University of Ulsan College of Medicine, Seoul, Republic of Korea, 7 Department of Urology, Center for Urologic Cancer, Hospital/Division of Tumor Immunology, Research Institute, National Cancer Center, Goyang, Republic of Korea

☉ These authors contributed equally to this work.
* seohk@ncc.re.k (HKS); bshong@amc.seoul.kr (BH)

## Abstract

### Background

Upper urinary tract urothelial carcinomas are relatively rare and have a cancer-specific survival rate of 20%–30%. The current gold standard treatment for nonmetastatic high-grade urinary tract urothelial carcinoma is radical nephroureterectomy with bladder cuff resection.

### Objective

This study aimed to compare conditional cancer-specific survival between open radical nephroureterectomy and laparoscopic radical nephroureterectomy in patients with nonmetastatic stage pT3-4 or TxN(+) locally advanced urinary tract urothelial carcinoma from five tertiary centers.

### Methods

The medical records of 723 patients were retrospectively reviewed. The patients had locally advanced and nodal staged tumors and had undergone open radical nephroureterectomy (n = 388) or laparoscopic radical nephroureterectomy (n = 260) at five tertiary Korean institutions from January 2000 and December 2012. To control for heterogenic baseline differences between the two modalities, propensity score matching and subgroup analysis were

**Data Availability Statement:** The data used in this study contain potentially identifying or sensitive patient information. The de-identified and anonymized data are available upon request from the IRB of the National Cancer Center Korea (NCC Korea) at "eirb@ncc.re.kr". All requests for the data containing potentially identifying or sensitive patient information will be examined by the IRB of the NCC Korea, and will only be made available after the approval of the IRB.

**Funding:** This study was supported by a Korean National Cancer Center grant (NCC1810242-1). However, the funder did not play any role in the study design, data collection, and analysis.

**Competing interests:** The authors have declared that no competing interests exist.

conducted. Conditional survival analysis was also conducted to determine survival outcome and to overcome differences in follow-up duration between the groups.

## Results

During the median 50.8-month follow up, 255 deaths occurred. In univariate analysis, significant factors affecting cancer-specific survival (e.g., age, history of bladder cancer, American Society of Anesthesiologists score, pathological N stage, and presence of lymphovascular invasion and carcinoma in situ) differed in each subsequent year. The cancer-specific survival between patients treated with open radical nephroureterectomy and laparoscopic radical nephroureterectomy was not different between patients with and without a history of bladder cancer. After adjusting baseline differences between the two groups by using propensity score matching, both groups still had no significant differences in cancer-specific survival.

## Conclusion

The two surgical modalities showed no significant differences in the 5-year cancer-specific survival in patients with locally advanced urinary tract urothelial carcinoma.

## Introduction

Upper urinary tract urothelial carcinomas (UTUCs) are relatively rare, and account for only 5% of urothelial tumors; they have a cancer-specific survival (CSS) rate of 20%–30% [1, 2]. Real world evidence regarding the prediction of CSS after surgery for UTUC, especially for Asian patients, is scarce [2, 3]. The current gold standard treatment of nonmetastatic, high-grade UTUC is radical nephroureterectomy (RNU) with bladder cuff resection [1].

Compared to open RNU (ONU), laparoscopic RNU (LNU) provides significantly better perioperative outcomes and comparable efficacy in terms of prognostic outcomes [4, 5]. However, the interpretation of data in patients with locally advanced UTUC is limited because patients managed with a laparoscopic approach are more likely to have Ta/Tis or T1 disease and less likely to have T3 or T4 lesion [4, 5]. Some reports have demonstrated that ONU was performed more often than LNU in patients with clinically nodal positive (N+) or nonmetastatic, high-grade, locally advanced UTUC [6]. Patients who undergo ONU are more likely to have more advanced (i.e., higher rate of N+) and more aggressive disease (i.e., higher rate of lymphovascular invasion [LVI]). In addition, ONU patients have longer follow-up times than do LNU patients, which results in a higher likelihood of observing CSS. Thus, bias may be introduced when determining the general survival estimate.

To minimize bias caused by these inherent baseline properties, our previous study adjusted several different parameters between the two groups to determine significant prognostic factors by using the propensity score matching technique [7]. In addition, to minimize bias caused by the inherent time data properties (e.g., different follow-up durations and the properties of a time-dependent changeable cohort that survived a certain period of time beyond treatment), the concept of conditional survival probability was introduced. The general survival rate is measured starting from the initial diagnosis of the disease [8]. By contrast, the CSS is calculated from the day of surgery to the most recent follow-up, however, this findings only reflects a static view of survival estimates and lacks postoperative follow-up information [9].

Hence, conditional CSS (CCSS) analysis is a good investigative method to estimate the likelihood of further survival and to compare the prognostic outcomes between groups with different follow-up durations [2, 10].

To the best of our knowledge, no concrete evidence exists in an actual clinical setting regarding the efficacy of open laparoscopic technique on CCSS, especially among patients with locally advanced UTUC in the Asian population. Some reports have demonstrated that LNU is not inferior to ONU and it has significantly comparable efficacy on survival, even in patients with locally advanced UTUC (pT3-4, pTxN+) [4, 11]. The aim of this study was to compare the CCSS between ONU and LNU in patients with advanced UTUC from five multicenters, after adjusting for the differential baseline characteristics by using propensity score matching.

## Materials and methods

### Ethics approval and consent to participate

The institutional review board of each multicenter institution approved this study, including the National Cancer Center (NCC-2016-0040 and 2018-0114-0001) [12]. The requirement for written informed consent was waived because of the retrospective design of the study. All patients' data and records were anonymized before the analysis.

### Patients' inclusion and exclusion criteria

We retrospectively reviewed the data of 723 patients with locally advanced UTUC (pT3, pT4, or pTxN+ without metastasis). Among these patients, 75 (10.4%) patients with a history of bladder cancer who underwent open RNU (ONU, *n* = 439, 60.7%) or laparoscopic RNU (LNU, *n* = 284, 39.3%) at five tertiary Korean institutions (National Cancer Center, Asan Medical Center, Samsung Medical Center, Seoul National University Hospital, and Korea University Hospital) for the Urothelial Cancer—Advanced Research and Treatment (UCART) Study conducted between January 2000 and December 2012 [11]. RNU cases before the year 2000 were excluded to eliminate the potential bias of surgical inexperience on LNU. The exclusion criteria were stage pTa-2N0M0 UTUC, history of or concomitant radical cystectomy or benign bladder surgery, bilateral UTUC, incomplete follow-up records, and history of neoadjuvant chemotherapy.

### Pathology, surgery, and follow-up

Tumor staging and grading were based on the 1998 World Health Organization/International Society of Urologic Pathology consensus classification for tumor grading and the 2010 American Joint Committee on Cancer/Union Internationale Contre le Cancer (Tumor-Node-Metastasis) classification for tumor staging [13, 14].

Based on a previously published paper of the original UCART data set,[12] the methodology of ONU or LNU and the postoperative follow-up protocols were not standardized but were performed at the surgeon's discretion. Unless the patient was contraindicated for laparoscopic surgery such as a history of abdominal surgery or high body mass indices, the LNU was indicated as the first surgical choice. Adjuvant chemotherapy was administered according to the pathological stage (stage pT3-4, N+). Postoperative follow-up was conducted from the year 2000 until the end of 2012, based on the follow-up regimen of the international UTUC guidelines [15].

## Statistical analysis

All analyzes were conducted on each of three data sets: the full dataset to use whole information included in the entire study population; a subgroup dataset to exclude relative vulnerability and consisting of patients with no bladder cancer history to eliminate its prognostic effect on survival outcomes in UTUC; and the matched dataset to control the heterogeneity of baseline characteristics between the two groups was formed by propensity score matching. The propensity score matching was conducted with all covariates. The propensity score was estimated from logistic regression model with all covariates, and two groups were matched one-to-one on the propensity score using the standard greedy matching algorithm. Model calibration procedures were conducted (Hosmer and Lemeshow test p = 0.908), and the discriminating ability was confirmed (AUC = 0.661). The descriptive statistics were presented as the median (minimum-maximum) for continuous variables and as the frequency (%) for categorical variables. Differences in baseline characteristics between the two surgical approaches were analyzed by using the chi-square test and Mann–Whitney $U$ test in the full and subgroup datasets, and by using Bowker's symmetry test and Wilcoxon signed-rank test for the matched dataset. The Cox proportional hazard regression model was used to estimate the effect size of each surgical approach as the predictor of an additional 5-year CCSS, given the 1- to 5-year survivorship. The marginal model approach was used to consider the dependency of the matched pair [16]. The covariates included in the multivariable model were selected by backward variable selection method.

The observed CSS probabilities were calculated by using the Kaplan–Meier estimator. The conditional survival estimates were derived by using the multiplicative law of probability: the conditional probability of surviving a further $t$ years, given that an individual has already survived $s$ years, is defined as $CS(t|s) = S(s+t)/S(s)$. The standard error of the conditional survival probability was estimated using Greenwood's formula. The difference between the two conditional survival probabilities was compared by using the $z$-test [17]. From the starting year of 2000 until the end of 2012, the patients were followed up and the duration was counted at the event time of CSS. If the CSS was not counted until the end of this study, then the case was censored at the time of the study. The results of the statistical tests were two-tailed, and $p<0.05$ was statistically significant. All analyses were conducted using Statistical Analysis System version 9.3 (SAS Institute, Inc., Cary, NC, USA) and R software version 3.3.3 (R Foundation for Statistical Computing, Vienna, Austria).

## Results

### Population characteristics

The characteristics of the 723 patients are in Table 1. The comparison between LNU and ONU in the full dataset revealed significant differences in age, the American Society of Anesthesiologists (ASA) score, tumor location, pathological N (pN) stage, and follow-up duration among the 723 patients ($p<0.05$, Table 1A). After eliminating patients with a history of bladder cancer known to eliminate the prognostic effect on survival (Table 1B), the ASA score and pN stage still significantly differed between ONU and LNU. However, 207 (47.2) patients received adjuvant chemotherapy and no group was insignificantly different (p>0.05, Table 1). To eliminate baseline heterogeneity between the two groups, propensity score matching was performed and resulted in nonsignificant differences, except in the follow-up duration and survival rate (Table 1C).

**Table 1. Comparison of baseline characteristics between open and laparoscopic radical nephroureterectomy in upper urinary tract urothelial carcinoma (A) among all patients (N = 723), (B) patients without previous history of bladder cancer only, and (C) after propensity-score matching.**

| Variables | (A) Full data set | | | (B) Subset without previous bladder cancer history | | | (C) Propensity Score-Matched data set† | | |
|---|---|---|---|---|---|---|---|---|---|
| | ONU | LNU | p-value | ONU | LNU | p-value | ONU | LNU | p-value |
| Total N | 439 | 284 | | 388 | 260 | | 260 | 260 | |
| Age, years | | | 0.037 | | | 0.055 | | | 0.495 |
| ≤ 65 | 211(48.1) | 159(56.0) | | 191(49.2) | 148(56.9) | | 143(55.0) | 136(52.3) | |
| > 65 | 228(51.9) | 125(44.0) | | 197(50.8) | 112(43.1) | | 117(45.0) | 124(47.7) | |
| Sex | | | 0.965 | | | 0.977 | | | 0.155 |
| Male | 316(72.0) | 204(71.8) | | 275(70.9) | 184(70.8) | | 190(73.1) | 188(72.3) | |
| Female | 123(28.0) | 80(28.2) | | 113(29.1) | 76(29.2) | | 70(26.9) | 72(27.7) | |
| BMI, kg/m² | 24.1(13.8–48.2) | 24.3(15.3–35.0) | 0.213 | 23.99(13.8–48.2) | 24.35(15.3–35.0) | 0.093 | 24.0(17.0–33.5) | 24.3(15.3–35.0) | |
| ASA score | | | 0.003 | | | 0.005 | | | 0.267 |
| 1 | 117(26.7) | 59(20.8) | | 103(26.6) | 56(21.5) | | 64(24.6) | 59(22.7) | |
| 2 | 276(62.9) | 211(74.3) | | 243(62.6) | 192(73.9) | | 181(69.6) | 187(71.9) | |
| 3 | 29(6.6) | 12(4.2) | | 27(7.0) | 10(3.9) | | 12(4.6) | 12(4.6) | |
| Unknown | 17(3.9) | 2(0.7) | | 15(3.9) | 2(0.8) | | 3(1.2) | 2(0.3) | |
| Previous bladder cancer, n(%) | | | 0.112 | | | | | | 0.790 |
| No | 347(79.0) | 242(85.2) | | - | - | | 219(84.2) | 218(83.9) | |
| Previous bladder tumor Hx. | 51(11.6) | 24(8.5) | | - | - | | 26(10.0) | 24(9.2) | |
| Concomitant bladder tumor Hx. | 41(9.3) | 18(6.3) | | - | - | | 15(5.8) | 18(6.9) | |
| Tumor location | | | 0.018 | | | 0.062 | | | 0.457 |
| Renal pelvis | 201(45.8) | 133(46.9) | | 182(46.9) | 126(48.5) | | 125(48.1) | 120(46.2) | |
| Ureter | 138(31.4) | 109(38.4) | | 127(32.7) | 99(38.1) | | 88(33.8) | 98(37.7) | |
| Both renal pelvis and ureter | 100(22.8) | 42(14.8) | | 79(20.4) | 35(13.5) | | 47(18.1) | 42(16.1) | |
| Tumor grade | | | 0.893 | | | 0.941 | | | 0.761 |
| Low grade | 41(9.3) | 28(9.9) | | 38(9.8) | 27(10.4) | | 25(9.6) | 25(9.6) | |
| High grade | 390(88.8) | 252(88.7) | | 343(88.4) | 229(88.1) | | 231(88.9) | 231(88.9) | |
| Unknown | 8(1.8) | 4(1.4) | | 7(1.8) | 4(1.5) | | 4(1.5) | 4(1.5) | |
| Pathological T stage | | | 0.106 | | | 0.213 | | | 0.739 |
| pTa/pT1-2 | 17(3.9) | 5(1.8) | | 14(3.6) | 5(1.9) | | 6(2.3) | 5(1.9) | |
| pT3-4 | 422(96.1) | 279(98.2) | | 374(96.4) | 255(98.1) | | 254(97.7) | 255(98.1) | |
| Pathological N stage | | | <0.001 | | | <0.001 | | | 0.558 |
| pNx | 217(49.4) | 125(44.0) | | 198(51.0) | 118(45.4) | | 128(49.2) | 125(48.1) | |
| pN0 | 123(28.0) | 128(45.1) | | 108(27.8) | 116(44.6) | | 103(39.6) | 104(40.0) | |
| pN1 | 99(22.6) | 31(10.9) | | 82(21.1) | 26(10) | | 29(11.2) | 31(11.9) | |
| Concomitant LVI | 184(41.9) | 99(34.9) | 0.058 | 158(40.72) | 89(34.23) | 0.095 | 88(33.8) | 96(36.9) | 0.441 |
| Concomitant CIS | 60(13.7) | 45(15.9) | 0.417 | 52(13.4) | 38(14.62) | 0.662 | 37(14.2) | 39(15.0) | 0.806 |
| Adjuvant CTx | 207(47.2) | 133(46.8) | 0.933 | 181(46.65) | 121(46.54) | 0.978 | 121(46.5) | 123(47.3) | 0.853 |
| Follow-up duration (months) | 58.7(0.1–180.0) | 40.7(0.7–125.0) | <0.001 | 61.2(0.1–180.0) | 42.9(0.7–125.0) | <0.001 | 61.7(0.1–180.0) | 40.3(0.7–125.0) | <0.001 |
| Death | 174(39.6) | 81(28.5) | 0.002 | 152(39.2) | 71(27.3) | 0.002 | 96(36.9) | 76(29.2) | 0.062 |

† using all variables as matching factors without follow-up duration because it is time to CSS.

ONU, open nephroureterectomy; LNU, laparoscopic nephroureterectomy; BMI, body mass index; ASA, American Society of Anesthesiologists.

## Significant risk factors of CSS

Table 2A shows the univariate analysis results of risk factors of CCSS during the 5-year follow-up after radical nephroureterectomy among all patients, the subset of patients without a history of bladder cancer (Table 2B), and patients after propensity score matching (Table 2C).

For the full dataset, significant factors at the baseline diagnostic year before RNU were age, history of bladder cancer, pN stage, and the presence of LVI and carcinoma in situ (CIS) ($p < 0.05$; Table 2A). At the postoperative first-year follow-up, the baseline and tumor grade were significant factors influencing first-year CCSS ($p < 0.05$). At the second-year follow-up after RNU, only the presence of CIS and baseline tumor location were significant factors for CCSS ($p < 0.05$). At the third-year follow-up, adjuvant chemotherapy was the only remaining factor favoring CSS after RNU ($p = 0.0452$).

In the matched dataset, many factors were significant for CCSS after RNU, compared to those in the full and subgroup datasets (Table 2C). This matching dataset demonstrated the explicit variation in the prognostic value of each risk factor more prominently than in the full and subgroup datasets.

## Comparison of the CCSS between the two surgical modalities

The comparison of CCSS between ONU and LNU showed marginal differences at the baseline time alone during the 5-year follow-up for full data set ($n = 732$; $p = 0.0461$) and for the subset cohort without previous bladder cancer ($n = 648$; $p = 0.0492$) (Table 3A and 3B). After adjusting for significant factors affecting CSS, no time point showed a significant difference ($p > 0.05$; Table 3A and 3B). Using the propensity score–matched cohort, the comparison of CCSS between ONU and LNU showed the same results of nonsignificant differences with/without adjustment for significant clinicopathological factors at certain time points ($p > 0.05$; Table 3C).

## CCSS curve

Kaplan–Meier curves and the CCSS probability curves over the additional duration (if a patient survived 1–5 years) for the full and matched datasets, based on the surgical approach and the 5-year conditional survival probabilities, are presented in Fig 1.

The CSS gradually decreased for each time point versus the baseline, regardless of the dataset.

Fig 2 shows the estimated 5- and 3-year conditional survival probabilities (survival rate, SE) based on tumor staging in the matched data set. The overall number of patients with locally advanced UTUC, including the pT3, pT4, and pN+ stages, showed nonsignificant differences in CSS between LNU and ONU in the 5-year conditional survival analysis ($p > 0.05$). In addition, no events were observed after 3 years so that the difference in conditional probability between the two groups could not be assessed (Fig 2).

## Discussion

This study's findings showed that the effects of both surgical techniques on CCSS were not significantly different, even after adjusting for baseline differences by using the matching methodology in locally advanced UTUC. One Korean study of 371 patients with locally advanced UTUC who underwent RNU reported that LNU had a significantly unfavorable 5-year CSS (LNU 66.1% vs. ONU 80.2%) and was an unfavorable predictor of overall survival (hazard ratio, 2.5) in pT3-4 stage UTUC ($p < 0.05$), whereas another study have shown other contradictory outcomes [6, 18]. However, a Japanese study by Abe et al [19]. of 83 patients with pT3-4

**Table 2. Univariate analysis of risk factors for conditional overall survival during 5-year follow-up after radical nephroureterectomy (A) among all patients (N = 723), patients without previous history of bladder cancer only (N = 648), and (C) after propensity-score matching.**

A) Full data set

| Variables | Baseline (HR, 95% CI) | 1 year (HR, 95%CI) | 2 year (HR, 95%CI) | 3 year (HR, 95%CI) | 4 year (HR, 95%CI) | 5 year (HR, 95%CI) |
|---|---|---|---|---|---|---|
| Total N | 723 | 663 | 573 | 522 | 502 | 489 |
| Age, years | | | | | | |
| > 65 vs. ≤ 65 | 1.486(1.149–1.923) | 1.498(1.121–2.004) | 1.198(0.805–1.781) | 1.250(0.718–2.179) | 1.564(0.780–3.136) | 2.142(0.869–5.278) |
| p-value | 0.0026 | 0.0064 | 0.3730 | 0.4301 | 0.2078 | 0.0979 |
| Sex | | | | | | |
| Female vs. Male | 0.906(0.676–1.213) | 0.929(0.670–1.289) | 0.672(0.415–1.088) | 0.745(0.390–1.422) | 0.790(0.355–1.760) | 0.618(0.205–1.864) |
| p-value | 0.5055 | 0.6587 | 0.1059 | 0.3717 | 0.5647 | 0.3932 |
| BMI, kg/m2 | 0.980(0.941–1.021) | 0.989(0.945–1.036) | 1.028(0.966–1.094) | 1.009(0.923–1.102) | 1.036(0.925–1.159) | 1.031(0.888–1.197) |
| p-value | 0.3417 | 0.6468 | 0.3864 | 0.8485 | 0.5435 | 0.6868 |
| ASA score | | | | | | |
| 2 vs. 1 | 1.163(0.852–1.587) | 1.092(0.777–1.535) | 1.074(0.680–1.697) | 0.966(0.520–1.795) | 1.355(0.583–3.145) | 1.419(0.467–4.314) |
| 3 vs. 1 | 1.063(0.580–1.951) | 1.143(0.604–2.160) | 0.851(0.326–2.224) | 0.624(0.142–2.749) | 0.608(0.075–4.943) | 1.053(0.117–9.442) |
| Unknown vs. 1 | 2.490(1.227–5.050) | 0.352(0.049–2.546) | 0.000(0.000->999.999) | 0.000(0.000->999.999) | 0.000(0.000->999.999) | 0.000(0.000->999.999) |
| p-value | 0.0900 | 0.6721 | 0.9581 | 0.9414 | 0.7966 | 0.9346 |
| Previous bladder cancer | | | | | | |
| Previous bladder vs. No | 1.888(1.288–2.767) | 1.520(0.931–2.483) | 1.023(0.446–2.343) | 1.468(0.526–4.100) | 0.545(0.074–4.010) | 0.837(0.112–6.282) |
| Concomitant bladder vs. No | 1.659(1.091–2.523) | 1.840(1.163–2.910) | 1.942(1.034–3.649) | 2.428(1.028–5.738) | 2.155(0.654–7.096) | 0.000(0.000->999.999) |
| p-value | 0.0008 | 0.0132 | 0.1181 | 0.1132 | 0.3599 | 0.9851 |
| Tumor location | | | | | | |
| Ureter vs. Renal pelvis | 1.143(0.854–1.530) | 1.272(0.919–1.760) | 1.724(1.113–2.670) | 1.691(0.894–3.197) | 2.185(0.970–4.919) | 1.867(0.627–5.555) |
| Both vs. Renal pelvis | 1.263(0.903–1.767) | 1.165(0.787–1.726) | 1.142(0.650–2.006) | 1.727(0.846–3.526) | 1.922(0.758–4.871) | 2.399(0.773–7.444) |
| p-value | 0.3664 | 0.3415 | 0.0419 | 0.1902 | 0.1500 | 0.2940 |
| Tumor grade | | | | | | |
| High grade vs. Low grade | 1.851(1.128–3.035) | 2.305(1.283–4.143) | 1.409(0.770–2.578) | 1.057(0.514–2.174) | 1.070(0.440–2.603) | 1.454(0.423–4.997) |
| Unknown vs. Low grade | 1.591(0.535–4.729) | 1.185(0.265–5.296) | 0.000(0.000->999.999) | 0.000(0.000->999.999) | 0.000(0.000->999.999) | 0.000(0.000->999.999) |
| p-value | 0.0500 | 0.0141 | 0.5388 | 0.9886 | 0.989 | 0.838 |
| Pathological T stage | | | | | | |
| pT3-4 vs. pTa/pT1-2 | 1.694(0.698–4.109) | 1.339(0.551–3.257) | 1.757(0.433–7.129) | >999.999(0.000->999.999) | >999.999(0.000->999.999) | >999.999(0.000->999.999) |
| p-value | 0.2436 | 0.5192 | 0.4300 | 0.9872 | 0.9902 | 0.9928 |
| Pathological N stage | | | | | | |
| N0 vs. Nx | 0.965(0.716–1.300) | 1.110(0.799–1.543) | 0.994(0.644–1.536) | 1.001(0.556–1.802) | 1.030(0.495–2.142) | 1.425(0.549–3.696) |
| N1 vs. Nx | 2.065(1.494–2.855) | 2.099(1.432–3.077) | 1.676(0.958–2.932) | 1.295(0.531–3.158) | 1.099(0.320–3.772) | 1.471(0.312–6.933) |
| p-value | < .0001 | 0.0004 | 0.1514 | 0.8385 | 0.8812 | 0.7404 |
| Concomitant LVI | | | | | | |
| Yes vs. No | 2.242(1.733–2.900) | 2.079(1.555–2.778) | 1.462(0.975–2.193) | 1.364(0.767–2.423) | 1.388(0.669–2.881) | 1.006(0.362–2.794) |
| p-value | < .0001 | < .0001 | 0.0663 | 0.2904 | 0.3787 | 0.9907 |
| Concomitant CIS | | | | | | |
| Yes vs. No | 1.453(1.047–2.018) | 1.721(1.203–2.461) | 1.989(1.226–3.227) | 1.963(0.954–4.041) | 1.084(0.330–3.562) | 0.000(0.000->999.999) |
| p-value | 0.0256 | 0.0030 | 0.0053 | 0.0671 | 0.8946 | 0.9920 |
| Adjuvant Chemotherapy | | | | | | |
| Yes vs. No | - | 1.233(0.922–1.649) | 1.479(0.994–2.201) | 1.841(1.049–3.231) | 1.579(0.785–3.177) | 0.895(0.360–2.228) |
| p-value | - | 0.1573 | 0.0538 | 0.0335 | 0.2001 | 0.812 |

B) Sub data set

| Variables | Baseline (HR, 95% CI) | 1 year(HR, 95%CI) | 2 year (HR, 95%CI) | 3 year (HR, 95%CI) | 4 year (HR, 95%CI) | 5 year (HR, 95%CI) |
|---|---|---|---|---|---|---|
| Total N | 648 | 601 | 523 | 475 | 458 | 445 |
| Age, years | | | | | | |

*(Continued)*

**Table 2.** (Continued)

| | | | | | | |
|---|---|---|---|---|---|---|
| > 65 vs. ≤ 65 | 1.425(1.082–1.878) | 1.421(1.047–1.929) | 1.275(0.847–1.919) | 1.269(0.711–2.266) | 1.507(0.742–3.063) | 2.002(0.793–5.052) |
| p-value | 0.0118 | 0.0241 | 0.2446 | 0.4199 | 0.2570 | 0.1418 |
| Sex | | | | | | |
| Female vs. Male | 0.848(0.620–1.160) | 0.888(0.630–1.251) | 0.681(0.419–1.108) | 0.780(0.405–1.502) | 0.775(0.347–1.734) | 0.623(0.205–1.894) |
| p-value | 0.3017 | 0.4971 | 0.1219 | 0.4570 | 0.5352 | 0.4045 |
| BMI, kg/m2 | 0.993(0.951–1.037) | 1.012(0.965–1.061) | 1.037(0.973–1.105) | 1.033(0.942–1.132) | 1.056(0.943–1.183) | 1.060(0.911–1.233) |
| p-value | 0.7620 | 0.6201 | 0.2627 | 0.4900 | 0.3421 | 0.4505 |
| ASA score | | | | | | |
| 2 vs. 1 | 1.186(0.849–1.656) | 1.102(0.771–1.576) | 1.076(0.673–1.720) | 0.892(0.474–1.676) | 1.363(0.584–3.177) | 1.403(0.457–4.306) |
| 3 vs. 1 | 1.033(0.535–1.994) | 1.142(0.587–2.221) | 0.868(0.331–2.276) | 0.602(0.137–2.649) | 0.588(0.072–4.783) | 1.023(0.114–9.166) |
| Unknown vs. 1 | 3.334(1.631–6.818) | 0.459(0.063–3.334) | 0.000(0.000->999.999) | 0.000(0.000->999.999) | 0.000(0.000->999.999) | 0.000(0.000->999.999) |
| p-value | 0.0110 | 0.7897 | 0.9652 | 0.9234 | 0.7810 | 0.9396 |
| Tumor location | | | | | | |
| Ureter vs. Renal pelvis | 1.139(0.835–1.553) | 1.232(0.878–1.730) | 1.643(1.050–2.571) | 1.578(0.813–3.062) | 2.116(0.940–4.765) | 1.797(0.604–5.346) |
| Both vs. Renal pelvis | 1.256(0.870–1.814) | 1.142(0.749–1.741) | 1.143(0.634–2.059) | 1.916(0.915–4.014) | 1.940(0.738–5.098) | 2.341(0.714–7.676) |
| p-value | 0.4438 | 0.4761 | 0.0842 | 0.1862 | 0.1694 | 0.3430 |
| Tumor grade | | | | | | |
| High grade vs. Low grade | 1.940(1.145–3.285) | 2.191(1.216–3.946) | 1.387(0.756–2.545) | 1.013(0.489–2.098) | 1.085(0.444–2.650) | 1.449(0.419–5.014) |
| Unknown vs. Low grade | 1.993(0.661–6.008) | 1.339(0.299–5.982) | 0.000(0.000->999.999) | 0.000(0.000->999.999) | 0.000(0.000->999.999) | 0.000(0.000->999.999) |
| p-value | 0.0475 | 0.0274 | 0.5726 | 0.9992 | 0.9840 | 0.8422 |
| Pathological T stage | | | | | | |
| pT3-4 vs. pTa/pT1-2 | 1.841(0.684–4.954) | 1.527(0.566–4.119) | 1.699(0.418–6.897) | >999.999(0.000->999.999) | >999.999(0.000->999.999) | >999.999(0.000->999.999) |
| p-value | 0.2268 | 0.4028 | 0.4586 | 0.9874 | 0.9901 | 0.9928 |
| Pathological N stage | | | | | | |
| N0 vs. Nx | 0.948(0.689–1.304) | 1.082(0.765–1.529) | 0.937(0.600–1.462) | 0.950(0.516–1.751) | 1.107(0.527–2.327) | 1.626(0.605–4.369) |
| N1 vs. Nx | 2.039(1.436–2.893) | 2.078(1.388–3.112) | 1.411(0.773–2.573) | 1.133(0.432–2.970) | 1.166(0.337–4.028) | 1.673(0.348–8.058) |
| p-value | <0.0001 | 0.0010 | 0.4223 | 0.9405 | 0.9500 | 0.5964 |
| Concomitant LVI | | | | | | |
| Yes vs. No | 2.050(1.557–2.701) | 1.890(1.391–2.566) | 1.407(0.924–2.141) | 1.410(0.777–2.559) | 1.438(0.689–3.004) | 1.076(0.383–3.019) |
| p-value | <0.0001 | <0.0001 | 0.1111 | 0.2578 | 0.3333 | 0.8897 |
| Concomitant CIS | | | | | | |
| Yes vs. No | 1.438(1.003–2.063) | 1.614(1.091–2.387) | 1.884(1.124–3.158) | 1.704(0.762–3.811) | 1.171(0.356–3.856) | 0.000(0.000->999.999) |
| p-value | 0.0482 | 0.0166 | 0.0162 | 0.1942 | 0.7954 | 0.9923 |
| Adjuvant Chemotherapy | | | | | | |
| Yes vs. No | - | 1.365(1.005–1.855) | 1.493(0.991–2.249) | 2.186(1.205–3.966) | 1.783(0.873–3.64) | 1.041(0.410–2.64) |
| p-value | - | 0.0466 | 0.0553 | 0.0100 | 0.1125 | 0.9328 |
| C) Matched data set | | | | | | |
| Variables | Baseline (HR, 95% CI) | 1 year (HR, 95%CI) | 2 year (HR, 95%CI) | 3 year (HR, 95%CI) | 4 year (HR, 95%CI) | 5 year (HR, 95%CI) |
| Total N | 520 | 487 | 424 | 391 | 373 | 363 |
| Age, years | | | | | | |
| > 65 vs. ≤ 65 | 1.486(1.074–2.057) | 1.538(1.081–2.187) | 1.3(0.826–2.045) | 1.557(0.855–2.838) | 2.161(0.981–4.76) | 2.540(0.922–6.999) |
| p-value | 0.0167 | 0.0167 | 0.2567 | 0.1479 | 0.0558 | 0.0715 |
| Sex | | | | | | |
| Female vs. Male | 0.882(0.612–1.271) | 0.888(0.597–1.319) | 0.732(0.411–1.303) | 0.917(0.467–1.801) | 1.291(0.558–2.987) | 1.047(0.329–3.330) |
| p-value | 0.5002 | 0.5554 | 0.2887 | 0.8020 | 0.5509 | 0.9377 |
| BMI, kg/m2 | 0.983(0.932–1.035) | 0.990(0.935–1.048) | 1.018(0.954–1.087) | 1.043(0.955–1.138) | 1.075(0.953–1.212) | 1.052(0.884–1.252) |
| p-value | 0.5099 | 0.7335 | 0.5893 | 0.3527 | 0.2400 | 0.5661 |
| ASA score | | | | | | |
| 2 vs. 1 | 1.379(0.912–2.086) | 1.383(0.89–2.151) | 1.361(0.791–2.34) | 1.487(0.706–3.13) | 4.001(0.993–16.117) | 2.140(0.500–9.155) |
| 3 vs. 1 | 1.217(0.543–2.727) | 1.443(0.609–3.42) | 0.707(0.16–3.129) | 0.652(0.077–5.481) | 0.000(0.000–0.000) | 0.000(0.000–0.000) |

(*Continued*)

**Table 2.** (Continued)

| | | | | | | |
|---|---|---|---|---|---|---|
| Unknown vs. 1 | 1.897(0.299–11.980) | 1.16(0.113–11.883) | 0.000(0.000–0.000) | 0.000(0.000–0.000) | 0.000(0.000–0.000) | 0.000(0.000–0.000) |
| p-value | 0.4612 | 0.5345 | < .0001 | < .0001 | < .0001 | < .0001 |
| **Previous bladder cancer** | | | | | | |
| Previous bladder vs. No | 1.627(0.998–2.655) | 1.538(0.852–2.777) | 1.296(0.502–3.343) | 1.581(0.437–5.72) | 0.000(0.000–0.000) | 0.000(0.000–0.000) |
| Concomitant bladder vs. No | 1.596(0.933–2.732) | 1.668(0.915–3.042) | 2.111(0.979–4.551) | 1.904(0.656–5.53) | 0.872(0.11–6.917) | 0.000(0.000–0.000) |
| p-value | 0.0590 | 0.1218 | 0.1538 | 0.4119 | < .0001 | < .0001 |
| **Tumor location** | | | | | | |
| Ureter vs. Renal pelvis | 1.154(0.821–1.621) | 1.328(0.922–1.912) | 2.013(1.235–3.282) | 2.29(1.154–4.547) | 2.851(1.138–7.143) | 2.628(0.756–9.138) |
| Both vs. Renal pelvis | 1.348(0.850–2.137) | 1.256(0.759–2.079) | 1.153(0.564–2.355) | 2.162(0.93–5.025) | 1.874(0.582–6.031) | 2.471(0.601–10.163) |
| p-value | 0.4116 | 0.2941 | 0.0127 | 0.0472 | 0.0815 | 0.2745 |
| **Tumor grade** | | | | | | |
| High grade vs. Low grade | 2.234(1.191–4.193) | 2.26(1.18–4.328) | 1.579(0.761–3.274) | 1.091(0.482–2.469) | 1.15(0.391–3.384) | 1.477(0.339–6.439) |
| Unknown vs. Low grade | 1.484(0.310–7.094) | 0.878(0.109–7.078) | 0.000(0.000–0.000) | 0.000(0.000–0.000) | 0.000(0.000–0.000) | 0.000(0.000–0.000) |
| p-value | 0.0407 | 0.0357 | < .0001 | < .0001 | < .0001 | < .0001 |
| **Pathological T stage** | | | | | | |
| pT3-4 vs. pTa/pT1-2 | 1.249(0.387–4.028) | 1.032(0.303–3.51) | 1.563(0.212–11.532) | >999.999(>999.999->999.999) | >999.999(>999.999->999.999) | >999.999(>999.999->999.999) |
| p-value | 0.7095 | 0.9604 | 0.6617 | < .0001 | < .0001 | < .0001 |
| **Pathological N stage** | | | | | | |
| N0 vs. Nx | 0.976(0.694–1.374) | 1.245(0.855–1.813) | 0.958(0.592–1.551) | 1.063(0.571–1.977) | 1.301(0.561–3.018) | 1.833(0.574–5.856) |
| N1 vs. Nx | 2.189(1.416–3.385) | 3.04(1.887–4.899) | 2.28(1.16–4.481) | 2.265(0.689–7.453) | 3.097(0.664–14.448) | 4.105(0.440–38.258) |
| p-value | 0.0006 | < .0001 | 0.0322 | 0.3976 | 0.3494 | 0.3776 |
| **Concomitant LVI** | | | | | | |
| Yes vs. No | 1.960(1.417–2.710) | 1.95(1.357–2.802) | 1.474(0.905–2.4) | 1.497(0.788–2.841) | 1.464(0.625–3.428) | 1.314(0.424–4.075) |
| p-value | < .0001 | 0.0003 | 0.1190 | 0.2176 | 0.3804 | 0.6365 |
| **Concomitant CIS** | | | | | | |
| Yes vs. No | 1.678(1.125–2.501) | 1.791(1.154–2.78) | 1.935(1.076–3.481) | 2.125(0.906–4.984) | 0.946(0.209–4.275) | 0.000(0.000–0.000) |
| p-value | 0.0111 | 0.0094 | 0.0276 | 0.0829 | 0.9420 | < .0001 |
| **Adjuvant Chemotherapy** | | | | | | |
| Yes vs. No | - | 1.619(1.139–2.301) | 2.045(1.279–3.27) | 2.194(1.165–4.132) | 1.972(0.875–4.444) | 0.978(0.320–2.995) |
| p-value | - | 0.0072 | 0.0028 | 0.0150 | 0.1016 | 0.9694 |

Numbers shown in Hazard ratio, (95% confidence interval).

ASA, American Society of Anesthesiologists; BMI, body mass index.

stage did not find any significant difference between LNU and ONU, which was similar to the findings of a large-numbered study. These results could be explained by the exclusion of early LNU cases, to account for urologists who had not yet overcome the learning curve of the laparoscopic procedure for UTUC. Owing to improved laparoscopic surgical technology, including retroperitoneal lymph node dissection, urologists are able to master the laparoscopic technique more quickly, easily perform nodal dissection, and patients have reduced surgical morbidity with early recovery and a shorter hospital stay without adverse effects on the prognostic outcomes [10, 12, 19–22].

Peyronnet et al [23]. conducted a systematic review of 42 studies involving 7554 patients, which included 2629 LNU patients. Most of these studies showed no significant difference in oncological outcome between LNU and ONU, and only three studies showed a significantly poorer CSS in LNU, especially in locally advanced pT3-4 stage UTUC. Peyronnet suggested that the laparoscopic bladder cuff was a major unfavorable influencing factor of LNU outcome.

**Table 3. Comparison of conditional cancer-specific survival between open and laparoscopy procedures during 5-year follow-up after radical nephroureterectomy (A) among all patients (N = 723), (B) patients without previous history of bladder cancer only (N = 648), and (C) after propensity-score matching.**

| Surgical type (LNU vs. ONU) | (A)Full data set | | | | | | | |
|---|---|---|---|---|---|---|---|---|
| | N | | Event | | Unadjusted | | Adjusted | |
| | ONU | LNU | ONU | LNU | HR(95% CI) | p-value | HR(95% CI) | p-value |
| Baseline | 439 | 284 | 156 | 78 | 0.758(0.578–0.995) | 0.046 | 0.900(0.680–1.190) | 0.459 |
| 1 year | 391 | 272 | 115 | 68 | 0.921(0.682–1.244) | 0.593 | 1.061(0.779–1.446) | 0.707 |
| 2 year | 335 | 238 | 64 | 35 | 0.942(0.622–1.426) | 0.777 | 0.889(0.586–1.350) | 0.582 |
| 3 year | 300 | 222 | 32 | 19 | 1.320(0.741–2.349) | 0.346 | 1.401(0.784–2.503) | 0.255 |
| 4 year | 293 | 209 | 26 | 6 | 0.637(0.260–1.563) | 0.325 | | |
| 5 year | 283 | 206 | 16 | 3 | 0.666(0.192–2.313) | 0.523 | | |
| **Surgical type (LNU vs. ONU)** | **(B)Subset without previous bladder cancer history** | | | | | | | |
| | N | | Event | | Unadjusted | | Adjusted | |
| | ONU | LNU | ONU | LNU | Unadjusted | p-value | Adjusted | p-value |
| Baseline | 388 | 260 | 135 | 68 | 0.746(0.557–0.999) | 0.049 | 0.853(0.632–1.150) | 0.297 |
| 1 year | 351 | 250 | 105 | 60 | 0.872(0.634–1.198) | 0.397 | 0.971(0.700–1.347) | 0.861 |
| 2 year | 302 | 221 | 61 | 32 | 0.888(0.577–1.367) | 0.591 | 0.866(0.562–1.334) | 0.515 |
| 3 year | 270 | 205 | 31 | 16 | 1.113(0.604–2.050) | 0.732 | 1.226(0.662–2.271) | 0.517 |
| 4 year | 263 | 195 | 25 | 6 | 0.632(0.257–1.552) | 0.317 | | |
| 5 year | 253 | 192 | 15 | 3 | 0.670(0.192–2.337) | 0.530 | | |
| **Surgical type (LNU vs. ONU)** | **(C)Propensity Score-Matched data set†** | | | | | | | |
| | N | | Event | | Unadjusted | | Adjusted | |
| | ONU | LNU | ONU | LNU | Unadjusted | p-value | Adjusted | p-value |
| Baseline | 260 | 260 | 84 | 73 | 0.935(0.682–1.284) | 0.680 | 0.933(0.674–1.292) | 0.677 |
| 1 year | 238 | 249 | 67 | 64 | 1.075(0.758–1.523) | 0.685 | 1.079(0.754–1.546) | 0.677 |
| 2 year | 208 | 216 | 40 | 32 | 1.012(0.614–1.667) | 0.963 | 0.998(0.607–1.641) | 0.994 |
| 3 year | 188 | 203 | 22 | 19 | 1.377(0.739–2.567) | 0.314 | 1.411(0.752–2.644) | 0.283 |
| 4 year | 183 | 190 | 17 | 6 | 0.686(0.263–1.786) | 0.440 | | |
| 5 year | 176 | 187 | 10 | 3 | 0.978(0.320–2.995) | 0.969 | | |

Adjusted for covariate selected from the backward selection.

Baseline—age, previous bladder cancer, pathological T and N stage, and concomitant lymphovascular invasion.

1-year—age, previous bladder cancer, tumor grade, pathological T and N stage, concomitant lymphovascular invasion.

2-year—tumor location and concomitant carcinoma in situ.

3-year—adjuvant chemotherapy.

4- and 5-year—not selected.

All results were described as hazard ratio (95% confidence interval).

ONU, open radical nephroureterectomy; LNU, laparoscopic radical nephroureterectomy.

However, this systematic review was based on studies conducted before August 2016. Recent systematic reviews have found contrary evidence that LNU had comparable oncological outcomes as that of ONU [4, 24, 25].

In the CCSS study by Ploussard et al., [2] 783 (22.1%) RNU patients had stage pT3, 140 (3.9%) patients had stage pT4, and 330 (9.3%) patient had stage pN+. Similar to the findings of our study (Table 2 and Fig 2), Ploussard showed that the probability of survival markedly increased over time in patients (the 1-year, 2-year, 3-year, 4-year and 5-year survival rates were 75.5%, 81.1%, 84.0%, 87.6%, and 88.8%, respectively) with high-stage disease. However, their aim was not to show the comparative efficacy between the two surgical RNU techniques in an advanced UTUC cohort; therefore, they did not consider certain inherent limitations of

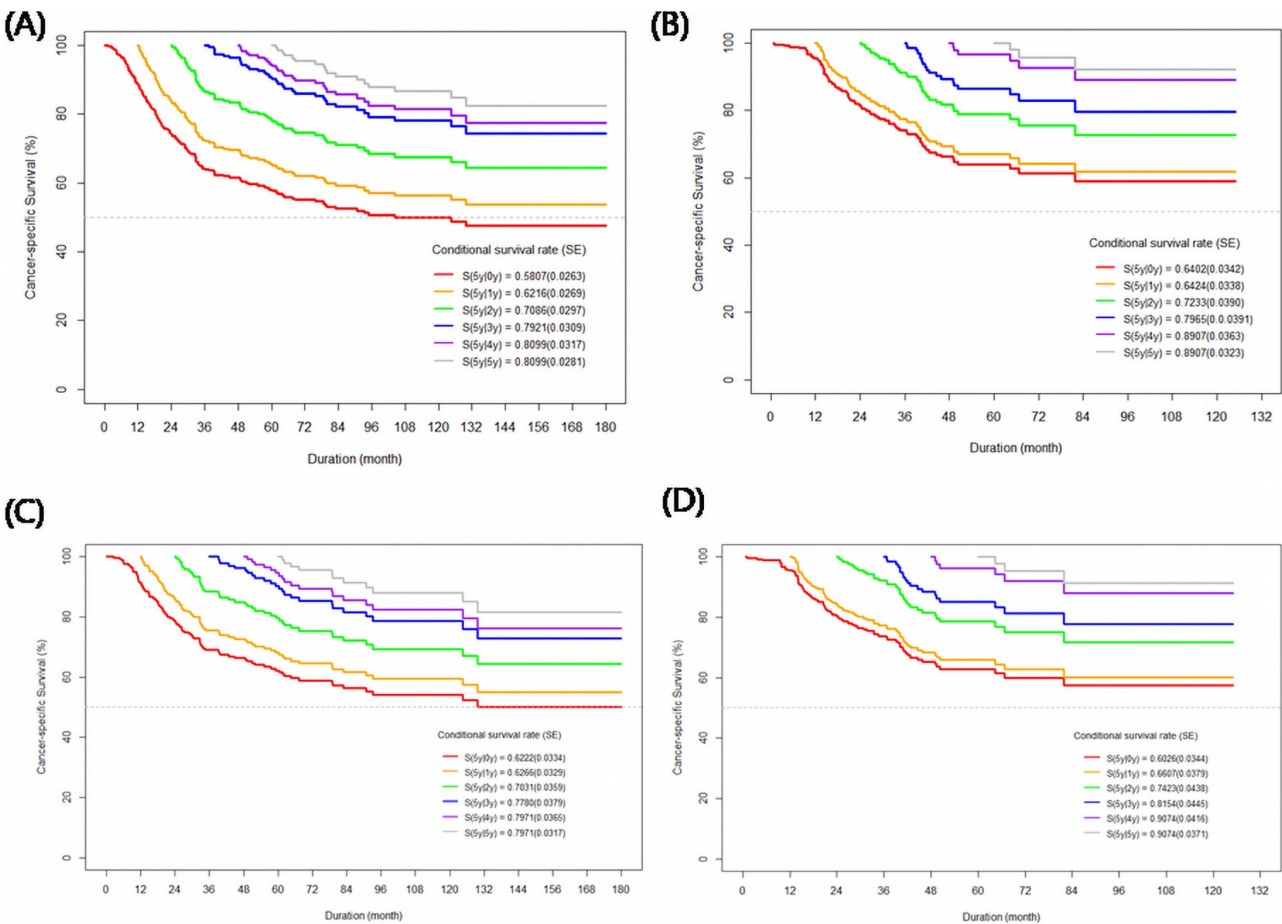

**Fig 1. Conditional cancer-specific survival curves according to the surgical approaches in full dataset (A, open vs B, laparoscopic RNU) and matched dataset (C, open vs B, laparoscopic RNU).** RNU, radical nephroureterectomy.

including early LNU cases, they did not adjust for baseline differences in the most important prognostic factors of locally advanced UTUC, and they did not control for the effect of the surgical learning curve and en bloc removal of a primary UTUC [24–28]. A study on CCSS in 146 patients with stage pT3-4 or pN+ by Kang et al [3]. demonstrated the prognostic outcomes of CCSS, based on the pathological stages, which were similar to our findings (p<0.05; Table 2). They showed the important role of LNU, which is comparable to ONU, in locally advanced UTUC, after adjusting for significant prognostic factors of CSS. However, their small number of patients was the main limitation (pT3, $n$ = 414, 42.7%; T4, $n$ = 5, 1.5%; and pN+, $n$ = 14, 4.2%).

This study showed the existence of different factors affecting a prognosis at each annual follow-up and the decreased impact of prognostic pathological features, which included LVI and pN stage and age, over time until their disappearance in long-term CCSS (Table 2), which were similar to the results of previous studies [2, 3]. In the study by Ploussard et al., age, adjuvant chemotherapy, surgical approach and all pathologic features were predictive factors for CSS, whereas sex and ureteral management type were not. However, some differences from previous studies [2, 3]. were observed in assessing these factors. In the studies by Ploussard et al.[2] and Kang et al.,[3] ASA score, pathological CIS, positive numbered nodal states, and

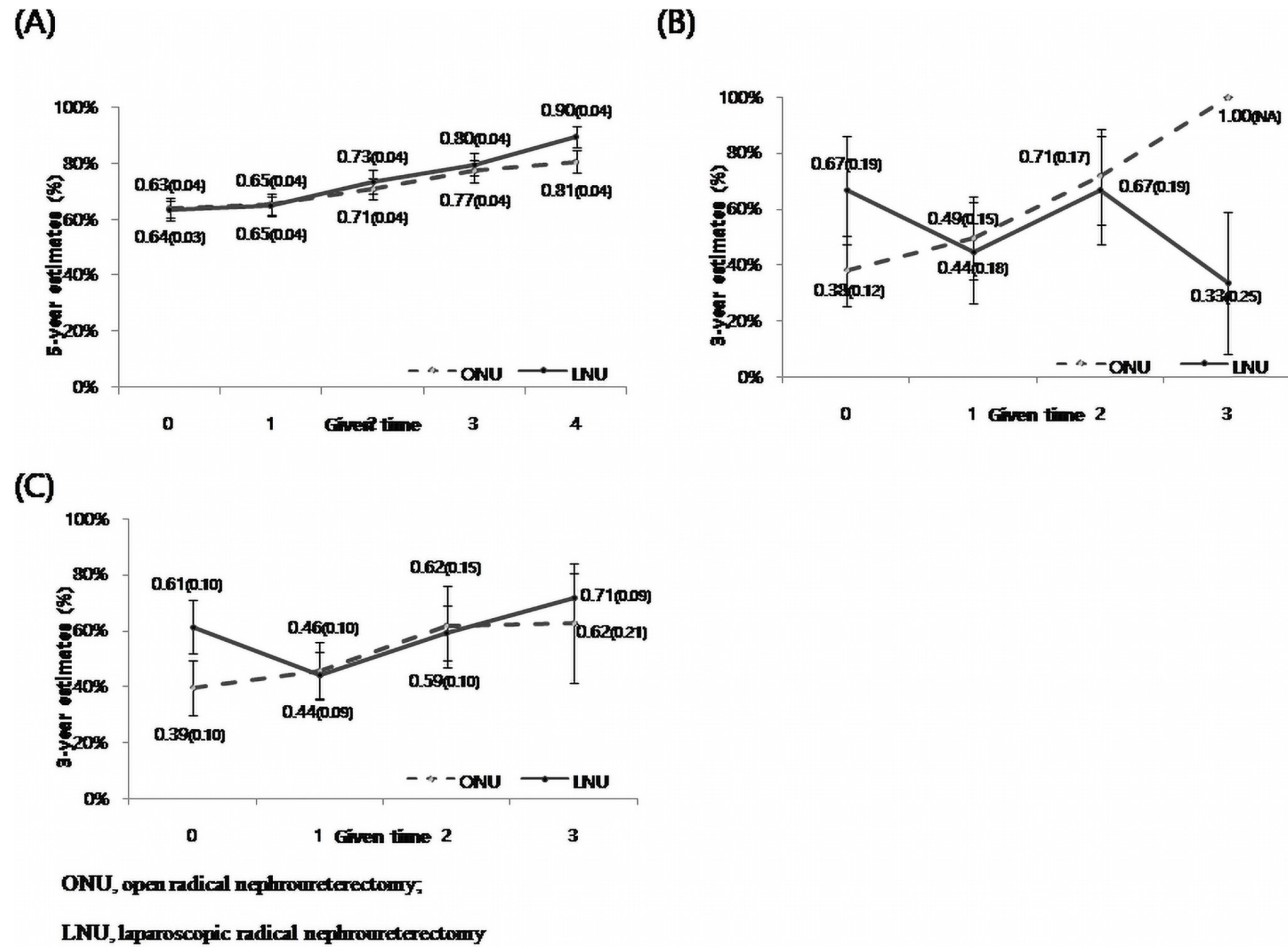

**Fig 2. Conditional probability of cancer-specific survival in stage pT3 (A), stage pT4 (B), and stage pN+ (C) upper urinary tract urothelial carcinoma groups according to the surgical approaches in the matched data set (N = 520).**

adjuvant chemotherapy were also significant prognostic factors affecting survival in the early years and in later years; however, in this study, pT stage and chemotherapy were not a significant factor for later years, even after propensity score matching (p>0.05, Table 2). In particular, a recent POUT clinical trial and meta-analytic report described the importance of adjuvant chemotherapy and its regimen and cycles for the prognoses of UTUC [26, 27]. All of these factors are clinically important when choosing an adequate surgical modality in high-risk patients with advanced UTUC.

Finally, a different naïve UTUC cohort without a bladder cancer history was analyzed because a history of bladder cancer is a well-known risk factor for cancer recurrence and invasiveness but not for locoregional recurrence/distant metastasis, CSS, or overall survival in UTUC [28–30]. Despite the significant differences in pN stage, ASA score, and follow-up duration between the two surgical groups (p<0.05, Table 2), the 5-year conditional survival analysis among naïve UTUC patients showed results nearly similar to those of patients with previous bladder cancer. An explanation for this finding may be that patients with a history of bladder cancer had more regular follow-ups with cystoscopy and upper tract evaluation, which led to an earlier diagnosis of UTUC and timely definitive management. Furthermore, this study showed that previous bladder cancer was a significant factor affecting the conditional

outcome of UTUC patients only until 1 year of postoperative follow-up, as revealed by the univariate analysis of the conditional survival outcome during the 5-year follow-up (Table 2). This finding indicated that a history of bladder cancer does not affect survival outcome, which is similar to previous data [28–30].

This study had several limitations, owing to its retrospective multicenter design, such as limited and different follow-up durations for each surgical group, and heterogeneity of surgical procedures and postoperative therapeutic decisions. To decrease the inherent bias because of the retrospective nature of baseline differences between the ONU and the LNU, additional propensity score matching was used with conditional survival analysis. However, a further prospectively designed study should be considered to eliminate these baseline biases, the selection bias of surgical indications, and the possibility of underestimated CCSS due to the dropout and censoring due to incomplete follow-up. This CCSS study gave better and more real-time informative estimates of outcome probability in clinical practice for follow-up and therapeutic strategy at each additional follow-up period.[6] A future prospective study with all the potential parameters of prognosis is necessary to improve the discriminatory ability of the predictive model for UTUC.

## Conclusions

This study showed that the 5-year CCSS was not significantly different between LNU and ONU in locally advanced UTUC. Propensity score matching, adjusted for the baseline differences of each group, showed a nonsignificant effect on CSS between the two surgical modalities. In addition, the conditional time showed that the importance of each significant prognostic parameter differed for early and late survival outcomes.

## Acknowledgments

All the following authors are members of the Urothelial Cancer–Advanced Research and Treatment (UCART) Study Group in Korea. We thank the UCART Study group and its leader Dr. Ho Kyung Seo for the clinical support.

Sung Han Kim and Ho Kyung Seo, from Department of Urology, Center for Urologic Cancer, National Cancer Center, Goyang, Republic of Korea.

Ja Hyeon Ku, from Department of Urology, Seoul National University Hospital, Seoul National University College of Medicine, Seoul, Republic of Korea.

Seok Ho Kang, from Department of Urology, Korea University Anam Hospital, Korea University College of Medicine, Seoul, Republic of Korea.

Byong Chang Jeong, from Department of Urology, Samsung Medical Center, Sungkyunkwan University School of Medicine, Seoul, Republic of Korea.

Bumsik Hong, from Department of Urology, Asan Medical Center, University of Ulsan College of Medicine, Seoul, Republic of Korea.

## Author Contributions

**Conceptualization:** Sung Han Kim, Mi Kyung Song, Ja Hyeon Ku, Byong Chang Jeong, Bumsik Hong, Ho Kyung Seo.

**Data curation:** Sung Han Kim, Mi Kyung Song, Ja Hyeon Ku, Seok Ho Kang, Byong Chang Jeong, Ho Kyung Seo.

**Formal analysis:** Mi Kyung Song.

**Funding acquisition:** Ho Kyung Seo.

**Investigation:** Sung Han Kim, Seok Ho Kang.

**Methodology:** Sung Han Kim, Seok Ho Kang.

**Project administration:** Bumsik Hong.

**Resources:** Ja Hyeon Ku, Seok Ho Kang, Byong Chang Jeong.

**Supervision:** Sung Han Kim, Mi Kyung Song, Seok Ho Kang, Byong Chang Jeong, Bumsik Hong, Ho Kyung Seo.

**Writing – original draft:** Sung Han Kim, Mi Kyung Song, Ho Kyung Seo.

**Writing – review & editing:** Sung Han Kim, Mi Kyung Song, Ja Hyeon Ku, Bumsik Hong, Ho Kyung Seo.

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
