## [Decision Letter · Decision Letter 0]

3 May 2021

PONE-D-21-04718

A retrospective multicenter analysis of conditional survival after radical nephroureterectomy for locally advanced upper tract urothelial carcinoma

PLOS ONE

Dear Dr. KIM,

Thank you for submitting your manuscript to PLOS ONE. After careful consideration, we feel that it has merit but does not fully meet PLOS ONE’s publication criteria as it currently stands. Therefore, we invite you to submit a revised version of the manuscript that addresses the points raised during the review process.

We look forward to receiving your revised manuscript.

Kind regards,

Isaac Yi Kim, MD, PhD

Academic Editor

PLOS ONE

Journal Requirements:

5. One of the noted authors is a group or consortium [UCART Study group]. In addition to naming the author group, please list the individual authors and affiliations within this group in the acknowledgments section of your manuscript. Please also indicate clearly a lead author for this group along with a contact email address.

Reviewers' comments:

Reviewer's Responses to Questions

**Comments to the Author**

1. Is the manuscript technically sound, and do the data support the conclusions?

Reviewer #1: Yes

Reviewer #2: Partly

2. Has the statistical analysis been performed appropriately and rigorously? 

Reviewer #1: I Don't Know

Reviewer #2: No

3. Have the authors made all data underlying the findings in their manuscript fully available?

Reviewer #1: No

Reviewer #2: Yes

4. Is the manuscript presented in an intelligible fashion and written in standard English?

Reviewer #1: Yes

Reviewer #2: Yes

5. Review Comments to the Author

Reviewer #1: In this manuscript, the authors examined comparing the conditional cancer-specific survival (CCSS) between open (ONU) and laparoscopic radical nephroureterectomy (LNU) in patients with locally advanced UTUC from five tertiary centers. They compared the surgical outcomes retrospectively with CCSS in 723 patients with locally advanced UTUC (pT3, pT4, and pN+ without metastasis) who underwent either 439 open RNU or 284 laparoscopic RNU at 5 institutions between 2000 and 2012. The concept of conditional survival is convincing when considering diseases associated with unfavorable prognosis at initial diagnosis or treatment, such as locally advanced UTUC, which suggested that the patients who have already survived for a certain duration after being diagnosed with cancer, will continue to survive for a specified duration.

In general, this report is interesting for the urologists to updated information on the surgical outcome for locally advanced UTUC. However, some points need to be cleared and to be described more in detail.

Major comments:

1. This study showed the existence of different factors affecting prognosis at each annual follow-up and the decreased impact of prognostic pathological features including pN stage and LVI over time until their disappearance in long-term CCSS. This result is similar to ones of previous studies. Ploussard G, et al. in European Urology, 2015 reported that the tumor location, the presence of carcinoma in situ, and the type of bladder cuff excision were continuously predictive for intravesical recurrence-free survival whatever the survivorship. The authors are encouraged to discuss any difference with this paper.

2.The previous study demonstrated the clinical characteristics such as the number of removed nodes, lymph node status (both number of removed nodes and number of positive nodes). Could the authors provide these additional clinical dates to make it clear if it the difference between the previous report or not?

3. Why patients with NAC were excluded in this study?

4. The operation time, bleeding amount, and the transfusion should be considered.

5. Clinical stage as well as pathological stage should be examined. If clinical stage was T3, would you prefer to laparoscopic surgery?

6. Details of AC should be discussed (number of courses, regimens)

7. The significance to demonstrate subset B data is not clear.

8. The method of propensity score matching should be described in detail.

Minor comments:

1.On Table3, number of events should be disclosed for laparo- or open surgery, respectively.

2. Fig2-A. Conditional probability of cancer-specific survival at 5 years should be described.

3. Fig.2-B. Was there significant difference in 3-year conditional probability of cancer-specific survival between LNU and ONU?

Reviewer #2: In this paper the authors performed a multi-center retrospective review of 723 patients who underwent either open or laparoscopic nephu for locally advanced upper tract urothelial carcinoma from 2000-2012. They compared survival between the two surgical approaches and then performed a propensity score matched conditional survival analysis to find that there was no difference in 5 year survival rates between open and laparoscopic approaches. Overall, this paper may be of interest given that it reports on a rarer urologic malignancy and in an Asian population. However, I have several concerns listed below.

Title:

- Doesn’t mention anything about surgical technique and this papers main objective is comparing survival after open vs laparoscopic RNU

Abstract:

- Correct line 51 to state that RNU is the gold-standard treatment for non-metastatic HIGH GRADE UTUC.

- Need to state your definition of locally advanced disease

- Are you measuring cancer specific survival or overall survival?

Introduction:

- Grammatical errors throughout

- Statement on line 80 that lap RNU provides better outcomes needs a reference

- I disagree with the statement on line 83 that most urologists believe ONU is superior to LNU. I think MIS is considered standard at this point and there is plenty of data already to support this. Focus on what makes your paper unique – i.e. multicenter, Asian population, statistical methods

- Again, are you looking at cancer specific or overall survival?

Methods:

- What are the indications for open or lap RNU at the institutions?

- Why did you exclude neoadjuvant chemotherapy? This is now considered standard of care at academic centers

- You state you included pT3 or N+ patients but results show the inclusion of pTa/T1-2 and N0

Results:

- What was the follow up and how complete? Has significant implications on the validity of conditional survival assessments…

- What variables were controlled for in the matching?

- The tables need better labels in the headings i.e. HR (95%CI). Also the tables switch between separate columns for p values in table 1 to separate rows in table 2. Stick with p values in a separate column.

- I think you need to look at more variables if you are doing a survival analysis and stage should be stratified differently as that is the most important predictor. pTa should not be in same group as pT2.

- No data on adjuvant chemotherapy in the results sections.

Discussion:

- There are more limitations that need to be addressed. The real limitation in retrospective series that compare open and MIS surgery is selection bias. I don’t think the propensity score matching can control for this.

- Talk about the limitations of conditional survival analysis. You need to have good long term follow up.

- Patients had surgery from 2000 to 2012. There have been a lot of recent advancements in the management of UTUC including learning curve of MIS, use of adjuvant chemo (POUT trial) and intravesical chemo at the time of surgery. All those factors make the current results not necessarily applicable to modern readers.

6. PLOS authors have the option to publish the peer review history of their article (what does this mean?). If published, this will include your full peer review and any attached files.

Reviewer #1: No

Reviewer #2: No

---

## [Author Response · Author response to Decision Letter 0]

18 Jun 2021

Authors’ Responses to the Reviewers’ Comments

Manuscript title: A retrospective multicenter analysis of conditional survival after radical nephroureterectomy for locally advanced upper tract urothelial carcinoma

Manuscript ID: PONE-D-21-04718

Responsesto Reviewer #1’sComments

Thank you for taking the time to reviewour manuscript and for providingthe following valuable comments.

Major comments:

Comment:

Comment 1. This study showed the existence of different factors affecting prognosis at each annual follow-up and the decreased impact of prognostic pathological features including pN stage and LVI over time until their disappearance in long-term CCSS. This result is similar to ones of previous studies. Ploussard G, et al. in European Urology, 2015 reported that the tumor location, the presence of carcinoma in situ, and the type of bladder cuff excision were continuously predictive for intravesical recurrence-free survival whatever the survivorship. The authors are encouraged to discuss any difference with this paper.

RESPONSE:We have added sentences in the Discussion section regarding the effects of significant variables found in the Ploussard et al. study such as all pathologic features, sex, age, adjuvant chemotherapy, surgical approach, and ureteral management type (page 10 lines 283-5andpage 11 lines 287-290).

Comment 2.The previous study demonstrated the clinical characteristics such as the number of removed nodes, lymph node status (both number of removed nodes and number of positive nodes). Could the authors provide these additional clinical dates to make it clear if it the difference between the previous report or not?

RESPONSE:We have addedtext regarding the importance of positive numbered nodal states in the Discussion section (page 10 line 287), whereas this study only stratified the nodal states by the pathological nodal staging. Data regarding specific numbers of positive LN unfortunately were not collected at the time of data enrollment.

Comment 3. Why patients with NAC were excluded in this study?

RESPONSE:Even though neoadjuvant chemotherapy (NAC) is the standard of care for muscle invasive bladder cancer, NAC has not been standardized before radical nephroureterectomy (NUx) in the upper urinary tract urothelial carcinoma (UUTC) due to the limitation of accurate staging. With these inherent limitation of NAC in UTUC, NAC is conducted by physicians’ subjective judgment. To decrease the confounding effect of the multicentric retrospective study with non-standardized NAC in UUTC, we excluded cases with NAC in this study.

Comment 4. The operation time, bleeding amount, and the transfusion should be considered.

RESPONSE: As previously reported prospective studies, which included 80 patients, have already shown insignificant differences in operative time with less bleeding amount with laparoscopic nephroureterectomy (NUx) than open NUx. These variables were not included in the analyses in this study.

Comment 5. Clinical stage as well as pathological stage should be examined. If clinical stage was T3, would you prefer to laparoscopic surgery?

RESPONSE: We agree with the reviewer’s remark about the importance of clinical stage in urothelial carcinoma (UTUC). However, theclinical staging of UTUChas many limitations. To date, the clinical staging of UTUC iscommonly used to identify low-risk UTUCto attempt nephron sparing.The purpose of this study was to compare the conditional cancer-specific survival between laparoscopic and open radical nephroureterectomy in locally advanced upper tract urothelial carcinoma. We chose pathologic staging in this study because the conditional survival started with the time point of nephroureterectomy (NUx). As for cT3 staged UTUC, laparoscopic surgery was preferred, if it were possible.

Comment 6. Details of AC should be discussed (number of courses, regimens)

RESPONSE:As the reviewer remarked, adjuvant chemotherapy is an important risk factor for UTUC Only the presence of adjuvant chemotherapy unfortunately was collected at the time of enrollment. No information about the specific number of cycles and regimens was collected. We have added the importance of adjuvant chemotherapy and its regimen and cycles in the Discussion section (page 11 lines 290-292), and added two new references, as follows:

#26.Birtle A, Johnson M, Chester J, Jones R, Dolling D, Bryan RT, et al.

Adjuvant chemotherapy in upper tract urothelial carcinoma (the POUT trial): a phase 3, open-label, randomised controlled trial.Lancet. 2020;395(10232):1268-1277

#27.Quhal F, Mori K, Sari Motlagh R, Laukhtina E, Pradere B, Rouprêt M, et al.Efficacy of neoadjuvant and adjuvant chemotherapy for localized and locally advanced upper tract urothelial carcinoma: a systematic review and meta-analysis. Int J Clin Oncol. 2020;25(6):1037-1054

Comment 7. The significance to demonstrate subset B data is not clear.

RESPONSE:Some researchers report that approximately 20% to 30% of patients with UTUC have a history of bladder cancer, anda history of NMIBC was independent risk factor for bladder recurrence. However, controversyexists regarding its relationship with cancer-specific survival (CSS) and overall survival (OS). Therefore, in this study, we added a dataset that excludes a history of bladder cancer, which can be a confounding variable in oncologic outcome. To clarify the purpose of eliminating ahistory of bladder cancer in subset B, we have added text in the Methods section (page 5 lines 157-159) and corrected the text in the Results section (page 6 lines 193-194).

Comment 8. The method of propensity score matching should be described in detail.

RESPONSE:We have added information about the methodology of propensity-score matching (page 5 lines 161-165).

Minor comments:

Comment 1.On Table3, number of events should be disclosed for laparo- or open surgery, respectively.

RESPONSE:We have added the number of events in the Table 3.

Comment 2. Fig2-A. Conditional probability of cancer-specific survival at 5 years should be described.

RESPONSE:We have added the survival rates in Figure 2.

Comment 3. Fig.2-B. Was there significant difference in 3-year conditional probability of cancer-specific survival between LNU and ONU?

RESPONSE:No significant difference existed in all 3-year conditional probabilities of cancer-specific survival between patients with LNU and patients with ONU who survived 0 to 2years. In ONU, which already has a survival of 3years, we could not estimate 3-year conditional probability of CSS because no events occurred after that. We also could not test the difference in conditional probability between two groups.We have added this information in the Results section (page 8 line 237-8).

Responses to Reviewer #2’sComments

Thank you for taking the time to review our manuscript and for providing the following valuable comments.

Comment:

Title:

- Doesn’t mention anything about surgical technique and this papers main objective is comparing survival after open vs laparoscopic RNU

RESPONSE:We have added some text in the title to describe the surgical methods (title page, lines 1-3).

Comment:Abstract: Correct line 51 to state that RNU is the gold-standard treatment for non-metastatic HIGH GRADE UTUC.

RESPONSE:We have corrected the sentence (page 1 line 53 and page 2 line 83).

Comment:Need to state your definition of locally advanced disease

RESPONSE:We have addedtext to indicate nonmetastatic, pT3-4 stage or LN(+), locally advanced UTUC (page 1 line 57 and page 2 line 92 and page 3 line 115).

Comment:Are you measuring cancer specific survival or overall survival?

RESPONSE:Our aim was to determine conditional cancer-specific survival (page 1 line 55 and page 3 line 115-118).

Comment:Introduction: Grammatical errors throughout

RESPONSE:We have asked anative English editor to correct the grammatical errors in the revised manuscript by using an English editing service (editage.com).

Comment:Statement on line 80 that lap RNU provides better outcomes needs a reference

RESPONSE:We have added references #4 and #5 in the manuscript (page 2 lines 87and90).

Comment:I disagree with the statement on line 83 that most urologists believe ONU is superior to LNU. I think MIS is considered standard at this point and there is plenty of data already to support this. Focus on what makes your paper unique – i.e. multicenter, Asian population, statistical methods

RESPONSE:We haverevised the sentence (page 2lines 90-92) and have added text to emphasize the uniqueness of this study using an Asian population (page 2 line 113, 116-117).

Comment:Again, are you looking at cancer specific or overall survival?

RESPONSE:Our aim was to determine conditional cancer-specific survival (page 1 line 53 and page 3 line 107).

Comment:Methods:What are the indications for open or lap RNU at the institutions?

RESPONSE: No specific indications were applied in this multicenter studies. If the case was not contraindicated for LNU such as a high body mass index or history of abdominal surgery, then LNU was indicated first. The most important factor in choosing a surgical technique is the judgement of the physicians, based on their own expertise and experience. We have added a sentence in the Methods section regarding the indicationsfor laparoscopic NUx(page 5 line 148-150).

Comment:Why did you exclude neoadjuvant chemotherapy? Today is now considered standard of care at academic centers

RESPONSE: We agree with the reviewer’s comment that the use of neoadjuvant chemotherapy (NAC) is increasing inacademic centers. However, to date, the use of NAC is not standardized before radical nephroureterectomy (NUx) for upper urinary tract urothelial carcinoma (UUTC),unlike bladder cancer.To decrease the confounding effect of the multicentric retrospective study with nonstandardized NAC in UUTC, we excluded patients with NAC in this study.

Comment:You state you included pT3 or N+ patients but results show the inclusion of pTa/T1-2 and N0

RESPONSE:We addeda small portion of patients with stages pTa-0 with a positivenode in this study. We changed the phrase regarding stage information(page 1 line 57 and page 3 line 115and page 4 line 129).

Comment:Results:What was the follow up and how complete? Has significant implications on the validity of conditional survival assessments…

RESPONSE:The follow-up guideline was in accordance with the follow-up regimen of the international UTUC guideline using regular cystoscopy, urinary markers, and imaging studies.

Beginning in year 2000 until the end of 2012, the patients were followed up and the duration was counted at the event time of cancer-specific survival. If cancer-specific survival was not counted until the end of this study, then the case was censored at the time of the study.We have added sentences to describe better the follow-up duration and the validity of conditional survival assessments in the Methods section (page 5 lines 151-153 and page 6 lines 180-183).

Comment:What variables were controlled for in the matching?

RESPONSE:We performed the propensity score matching with all covariates. Therefore, no significance existed in the whole variablesin Table1, except for the follow-up duration and survival rate. We tried to control heterogeneity through the propensity score matching, although the effect of the covariates on the outcome mayhave remain. Therefore, we further used theCox proportional hazard regression model with some covariates selected by the backward method. Please refer to “Adamina M. et al., 2006; Gum P. A. et al., 2001” for information about this method (page 5 line 161-165).

Comment:The tables need better labels in the headings i.e. HR (95%CI). Also the tables switch between separate columns for p values in table 1 to separate rows in table 2. Stick with p values in a separate column.

RESPONSE:We have added subheadings for HR (95% CI ) in Table 2. We also changed the Table 1 so that the p-value is in a separate column.

Comment:I think you need to look at more variables if you are doing a survival analysis and stage should be stratified differently as that is the most important predictor. pTa should not be in same group as pT2.

RESPONSE:We agree with the reviewer’s remark presenting differentially stratified stages of pTa from T2 stage. ThepTaN+ and pT1N+ groups were added tothe pT2 group because of the small number of patients (i.e., only one patient withpTaN+ stage and three patients with pT1N+ stage). Furthermore, our aim in this study was to examine the CCSS for patients withpT3–pT4, compared to that of patients without pT3–pT4.

T stage Full data set Subgroup data set Matched data set

 ONU LNU ONU LNU ONU LNU

pTa 1 (0.23) 0 (0.00) 1 (0.26) 0 (0.00) 0 (0.00) 0 (0.00)

pT1 3 (0.68) 0 (0.00) 3 (0.77) 0 (0.00) 2 (0.77) 0 (0.00)

pT2 13 (2.96) 5 (1.76) 10 (2.58) 5 (1.92) 4 (1.54) 5 (1.92)

pT3 394 (89.75) 272 (95.77) 348 (89.69) 249 (95.77) 237 (91.15) 248 (95.38)

pT4 28 (6.38) 7 (2.46) 26 (6.70) 6 (2.31) 17 (6.54) 7 (2.69)

Comment:No data on adjuvant chemotherapy in the results sections.

RESPONSE:We have added a sentence regarding adjuvant therapy in the Results section (page 7 lines 195-197).

Comment:Discussion:There are more limitations that need to be addressed. The real limitation in retrospective series that compare open and MIS surgery is selection bias. I don’t think the propensity score matching can control for this.

RESPONSE:We agree with the reviewer’s remark about the presence of selection bias for the indication of surgical approaches, despitehaving conducted propensity score matching. However, the clinical importance of this study is the scarcity of CCSS in UTUC in locally advanced stages in the Asian population. The results revealed some significant findingsin this study. We have addedtext regarding the limitation of selection bias and we have addedasuggestion regarding the necessity of using a prospectively designed study to eliminate inherent selection biases of the two surgical groups (page 11 line 311- page 12 lines 316).

Comment:Talk about the limitations of conditional survival analysis. You need to have good long term follow up.

RESPONSE:There is the limitation of not meeting the prerequisite that should be an almost complete followup in all patients. Thus, because of factors such as dropout, censoring occurred in the middle of the followup, and the conditional survival rate may have been underestimated.We have added this limitation of CCSS in the Discussion section (page 12 lines 315-316).

Comment:Patients had surgery from 2000 to 2012. There have been a lot of recent advancements in the management of UTUC including learning curve of MIS, use of adjuvant chemo (POUT trial) and intravesical chemo at the time of surgery. All those factors make the current results not necessarily applicable to modern readers.

RESPONSE:We understand the concerns raised in the reviewer’s remarks. Weattempted to make surgical skills equivalent by eliminating the learning curve effect of laparoscopic surgery and enrolling patientsafter 2000. In addition, adjuvant chemotherapy was added in the analysis to analyze the conditional cancer-specific survival between laparoscopic and open radical nephroureterectomy in locally advanced upper tract urothelial carcinoma.Intravesical chemotherapyis not covered by the national insurance in Korea, and it is used in limited cases with specifically selected purposes such as academic objectives.

---

## [Editor Report · Decision Letter 1]

28 Jul 2021

A retrospective multicenter comparison of conditional cancer-specific survival between laparoscopic and open radical nephroureterectomy in locally advanced upper tract urothelial carcinoma

PONE-D-21-04718R1

Dear Dr. Seo,

We’re pleased to inform you that your manuscript has been judged scientifically suitable for publication and will be formally accepted for publication once it meets all outstanding technical requirements.

Kind regards,

Isaac Yi Kim, MD, PhD

Academic Editor

PLOS ONE
---

## [Editor Report · Acceptance letter]

27 Sep 2021

PONE-D-21-04718R1 

A retrospective multicenter comparison of conditional cancer-specific survival between laparoscopic and open radical nephroureterectomy in locally advanced upper tract urothelial carcinoma 

Dear Dr. Seo:

I'm pleased to inform you that your manuscript has been deemed suitable for publication in PLOS ONE. Congratulations! Your manuscript is now with our production department. 

Kind regards, 

on behalf of

Dr. Isaac Yi Kim 

Academic Editor

PLOS ONE